# Martensitic Transition and Superelasticity of Ordered Heat Treatment Ni-Mn-Ga-Fe Microwires

Yanfen Liu [1,*], Zirui Lang [1], Hongxian Shen [2], Jingshun Liu [3] and Jianfei Sun [2]

1   Department of Physics, Qiqihar University, Qiqihar 161006, China
2   School of Materials Science and Engineering, Harbin Institute of Technology, Harbin 150001, China
3   School of Materials Science and Engineering, Inner Mongolia University of Technology, Hohhot 010051, China
*   Correspondence: lxylyf_0@163.com

**Abstract:** The preparation of Ni-Mn-Ga and Ni-Mn-Ga-Fe master alloy ingots and microwires was completed by high vacuum electric furnace melt melting furnace and melt drawing liquid forming equipment, and the lattice dislocations and defects formed inside the microwires during the preparation process were corrected by stepwise ordered heat treatment. The micro-structure and phase structure were characterized using a SEM field emission scanning electron microscopy and an XRD diffractometer combined with an EDS energy spectrum analyzer; the martensitic phase transformation process of the microwires was analyzed using a DSC differential scanning calorimeter; and the superelasticity of the microwires was tested by a Q800 dynamic mechanical analyzer. The results indicate that Fe doping can refine the grain, transform the phase structure from parent phase to single 7M martensite, reduce the number of martensitic variants, and increase the mobility of the twin grain boundary interface. The MT phase transition temperature ($M_S$) is substantially increased in the martensite transition (MT) process by the increase of the number of free electrons in its lattice. During the superelasticity (SE) test, both microwires displayed superior recover-ability of SE curves, and the Fe doping curves showed similar characteristics of "linear superelasticity", showing higher critical stress values and complete SE in the experiment. The critical stress satisfies the Clausius-Clapeyron equation and exhibits higher temperature sensitivity than Ni-Mn-Ga microwires.

**Keywords:** ferromagnetic shape memory alloy microwires; ordered heat treatment; martensite transformation; superelasticity





## 1. Introduction

Shape memory alloy (SMA) is one of the current hot spots in mechanosensitive materials research [1] because of its own characteristics of being sensed and driven simultaneously. Among them, ferromagnetic shape memory alloy (FSMA) has the characteristics of applied magnetic field and temperature modulated SMA martensite phase transition temperature (MT) [1–3]. A certain macroscopic shear can be observed by magnetically controlled martensite modification rearrangement which can enhance the response efficiency of conventional SMA and increase the application of FSMA [4]. Since the discovery of the excellent magnetic properties of $Ni_2MnGa$ single-crystal [5,6] SMA by Ullakko in 1996, the research on Ni-Mn-Ga SMA has been predominantly focused on single crystals. Yet, due to the high fabrication cost, difficult fabrication process [5,7], and low reproducibility of the experiments, the research on SMA has been carried out in the direction of "polycrystalline" [8]. However, size, inertia, and resistance were found to the MT temperature, SE training, and twinning deformation when the experimentally selected materials were studied in the "bulk polycrystalline" direction [9–11], so the materials were selected to transition to small-size materials [4]. In this experiment, small-size microwires obtained from bulk polycrystals prepared by melt drawing [4,12,13] were used to investigate the micro-structure, martensitic phase transformation and SE properties of FSMA microwires.

Meanwhile, in this experiment, based on the defects of low MT temperature and high intrinsic brittleness exhibited by the Ni-Mn-Ga polycrystalline alloy microwires, a fourth doping element was selected to strengthen them. It is known from the literature that the fourth element added is Cu, Mn, Co, Nd [14–16], which enhanced MT and microstructure in the experiments. Fe, the fourth group element, was chosen in this experiment because it is an austenitic stabilizing element with the characteristic skills of high temperature resistance, solid solution strengthening, microwire grain refinement; the ordered heat-treated Ni-Mn-Ga and Ni-Mn-Ga-Fe microwires were subjected to comparative experiments, in which they exhibited enhanced *Ms*, increased enthalpy change Δ*H*, broadened SE interval and increased microwire toughness.

## 2. Experimental Procedure

The 99.99% high-purity Ni, Mn, Ga, and Fe were selected for the preparation of Ni-Mn-Ga and Ni-Mn-Ga-Fe poly-crystalline alloy microwires. The selected experimental apparatus was a magnetically controlled tungsten arc furnace and a diamond cutting machine to obtain 8~10 mm ingots and 3~5 mm rods of the master alloys, respectively; $20 \pm 2\mu m$ microwires without obvious defects were prepared using the melt pulling device [4,12,13] in Figure 1a. The manufactured microwires were subjected to the stepwise ordering heat treatment of Figure 1b to obtain microwires of FSMAs with periodic atomic arrangement.

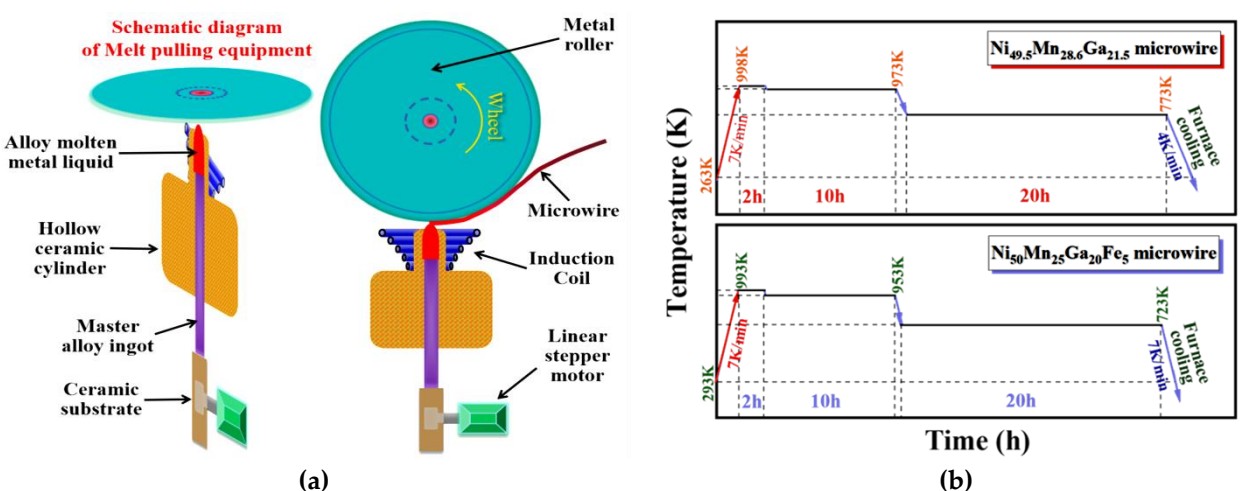

**Figure 1.** (**a**) Principle diagram of melt drawing equipment, (**b**) ordered heat treatment curve.

The ordered heat-treated microwires were observed by XRD diffractometer, SEM scanning electron microscope combined with an EDS energy spectrum analyzer to observe the phase and micro-structure of the microwires; the MT temperature of the microwires was calibrated by a DSC differential calorimetry scanner; the superelasticity (SE) of the microwires was tested by a Q800 dynamic mechanical analyzer [13,17]. The temperature at which the SE test was performed satisfied $T_{test} > Ms$, and the rate of temperature rise and fall was 5 K/min. Before the experiment, the microwires were preloaded with 0.01N to ensure that they were in an elastic state, and the load size was determined by the critical stress values of the two microwires, and the rates of loading and unloading were 0.02 N/min and 0.04 N/min, respectively.

## 3. Results and Discussion

Figure 2 illustrates the surface and cross-sectional structure of the ordered heat-treated microwires under SEM scanning electron microscopy; (a) is the surface diagram of Ni-Mn-Ga microwire, in which A, B indicate the flat surface and round surface of the microwire, respectively; the formation of this structure is due to the automatic formation of a circular surface by gravity in the uncontacted part of the copper wheel during the drawing process,

which coincides with the "D" structure of the fracture cross section of Ni-Mn-Ga microwire in (c). (b) and (d) demonstrate the circular surfaces and fracture cross-sections of Ni-Mn-Ga-Fe microwires. The Ni-Mn-Ga-Fe microwire grain size $l$ =1.264 ± 0.195 μm, which is smaller than that of Ni-Mn-Ga microwire grain $l$ = 2.526 ± 0.150 μm, when comparing Figure 2a, b, c and d in the figure respectively, indicates that Fe doping can refine the microwire grains; the twin width of the fracture surface is substantially diminished, the twin boundary is more obvious, and the twin is shaped from the preferential nucleation region toward the direction of the arrow in Figure 2d. The equations of the two microwire electron concentrations $e/a$ [18] were obtained by combining the EDS energy spectrum analysis of Table 1.

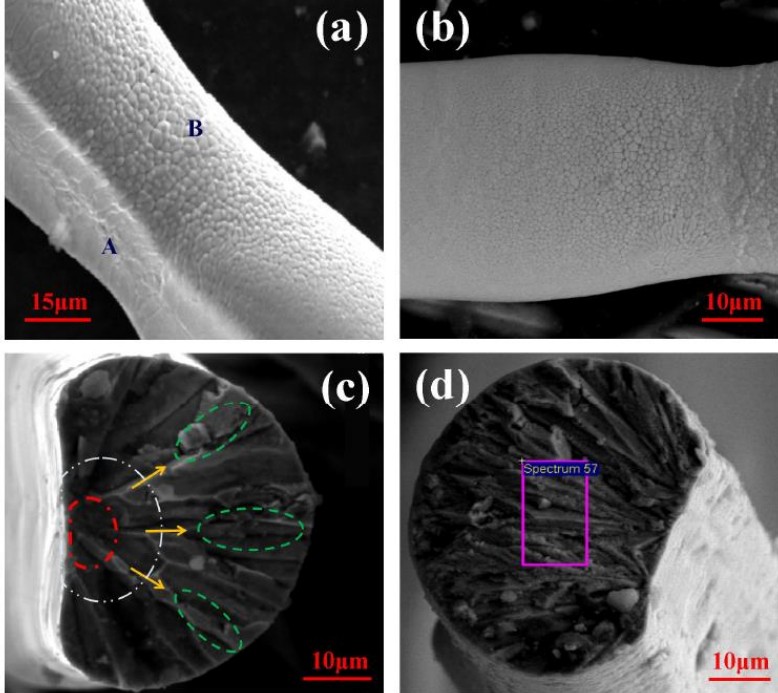

**Figure 2.** (**a**) Surface morphology of ordered heat-treated Ni-Mn-Ga microwires, (**b**) circular surface of ordered heat-treated Ni-Mn-Ga-Fe microwires, (**c**) fracture cross section of ordered heat-treated Ni-Mn-Ga microwires, (**d**) fracture cross section of ordered heat-treated Ni-Mn-Ga-Fe microwires.

**Table 1.** Results of the EDS energy spectrum analyzer for the detection of two alloy microwires.

| Ni-Mn-Ga Polycrystalline Alloy Microwires after Ordered Heat Treatment | | | | Ni-Mn-Ga-Fe Polycrystalline Alloy Microwires after Ordered HEAT Treatment | | | | |
|---|---|---|---|---|---|---|---|---|
| Element Atomic % | | | $e/a$ | Element Atomic % | | | | $e/a$ |
| Ni | Mn | Ga | | Ni | Mn | Ga | Fe | |
| 49.926 | 28.566 | 21.508 | 7.637 | 50.007 | 25.055 | 19.920 | 5.018 | 7.754 |
| 50.125 | 28.501 | 21.374 | 7.649 | 50.573 | 24.299 | 19.605 | 5.523 | 7.788 |
| 50.402 | 29.002 | 20.596 | 7.688 | 51.266 | 22.447 | 20.870 | 5.417 | 7.757 |
| 50.999 | 25.037 | 23.964 | 7.571 | 51.709 | 22.611 | 20.503 | 5.177 | 7.783 |
| 52.570 | 23.644 | 23.786 | 7.626 | 52.003 | 21.900 | 20.499 | 5.598 | 7.796 |

For ordered heat-treated Ni-Mn-Ga alloy microwires there is the formula:

$$e/a = (\text{Ni})\text{at.\%} \times 10 + (\text{Mn})\text{at.\%} \times 7 + (\text{Ga})\text{at.\%} \times 3 \tag{1}$$

For ordered heat-treated Ni-Mn-Ga-Fe alloy microwires there is the formula:

$$e/a = (\text{Ni})\text{at.}\% \times 10 + (\text{Mn})\text{at.}\% \times 7 + (\text{Ga})\text{at.}\% \times 3 + (\text{Fe})\text{at.}\% \times 8 \qquad (2)$$

The calculation of *e/a* in Table 1 shows that *e/a* > 7.7 for Ni-Mn-Ga-Fe microwires and 7.5 < *e/a* < 7.7 for Ni-Mn-Ga microwires, the theoretical analysis suggests that Ni-Mn-Ga microwires become a mixed state of 5M and 7M modulated martensite at MT [19,20], while Ni-Mn-Ga-Fe microwires become a single state of 7M modulated martensite at MT [19,20]. To verify the accuracy of the theoretical analysis, the microwires were phase-probed by XRD diffractometer, and the selected probing temperatures were 298K and 270K for the austenite and martensite states of $Ni_{49.9}Mn_{28.6}Ga_{21.5}$ microwires, and 345 K and 335 K for the austenite and martensite states of $Ni_{50}Mn_{25}Ga_{20}Fe_5$ microwires, respectively. As shown in Figure 3a, the austenitic states of both microwires are the same, A(220), A(400) and A(422), indicating that Fe doping did not affect the lattice structure of Ni-Mn-Ga microwires; at the martensitic state, the main peaks A(220) and A(422) of austenite of $Ni_{50}Mn_{29}Ga_{21}$ microwires split into mixed austenite and martensite diffraction peaks of A(220), M(220), M(202) and A(422), M(204), M(224); the main peaks A(220) and A(422) of $Ni_{50}Mn_{25}Ga_{20}Fe_5$ microwire austenite split into M(202), M(220), M(022) and M(242), M(422), M(224) complete martensitic diffraction peaks. From the two figures, it can be seen that the $Ni_{50}Mn_{29}Ga_{21}$ microwire phase diagram is in a mixed state of austenite and martensite, indicating that the MT is not complete; on the contrary, the $Ni_{50}Mn_{25}Ga_{20}Fe_5$ microwire diagram is in a completely martensitic state. Combined with the lattice parameter calculation, it was found that the $Ni_{50}Mn_{29}Ga_{21}$ microwire satisfies the 5M martensite structure with $a = b \neq c$ [21] ($a = 0.5829$ nm, $b = 0.5829$ nm, $c = 0.5641$ nm), while the $Ni_{50}Mn_{25}Ga_{20}Fe_5$ microwire satisfies the $a > b > c$ [21] ($a = 0.6065$ nm, $b = 0.5782$ nm, $c = 0.5534$ nm) structure, forming an orthogonal 7M martensite structure with a more regular internal atomic arrangement.

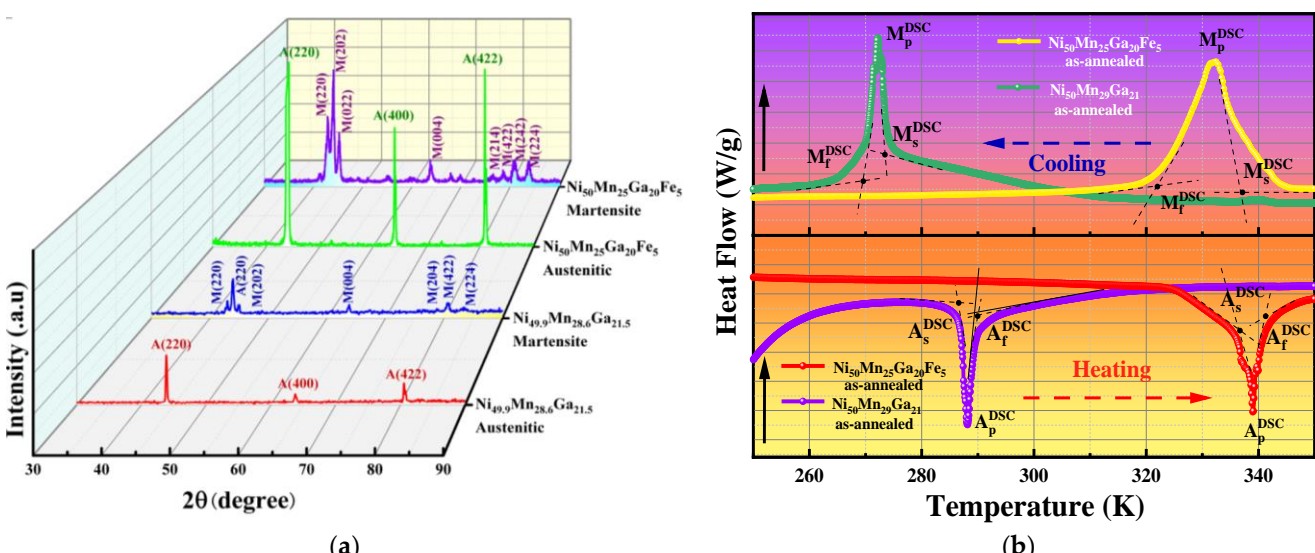

**Figure 3.** (**a**) XRD diagram of polycrystalline alloy microwire, (**b**) DSC diagram of polycrystalline alloy microwire.

From the above, it can be seen that Fe doping changes the micro-structure of Ni-Mn-Ga microwires in the martensitic state, which is mainly attributed to the increase of electron concentration and the decrease of the atomic radius. Because the atomic radius of Fe is smaller than that of Ga ($r_{Fe} = 0.172$ nm < $r_{Ga} = 0.181$ nm), Fe doping changes the first nearest neighbor of the internal lattice of the microwire from Ga-Ga to Ga-Fe, reduces the lattice volume, compacts the lattice arrangement within the microwire, and is able to transform directly from the austenitic state to the 7M martensitic state when stress is applied; it reduces the formation of 5M martensite under low stress conditions, due to

the work done by $Ni_{50}Mn_{29}Ga_{21}$ microwires in MT to overcome the twin boundary and the need for the process of releasing the internal driving force of 5M martensite to 7M martensite under conditions of increasing stress load and temperature. Meanwhile, the increase of electron concentration is due to the valence electron number $3d^6 4s^2$ of Fe than $4s^2 4p^1$ of Ga, the outermost free electron energy level is in the excited state; according to the bubbly principle, the free electron of Fe behaves more actively when the leap occurs, the doping makes the number of degrees of the microwire freedom increase, and the dense surface structure formed by the free particles during microwire formation is more dense and easier to obtain 7M martensite.

The increase in $e/a$ is one of the influencing factors for the MT temperature enhancing in addition to affecting the lattice structure. Figure 3b illustrates the DSC calorimetric curves of the microwires, and Table 2 shows the measured data of the DSC calorimetric curves. As can be seen from the figure, Fe doping shifts $M_S$ to the high temperature side, indicating that the martensite phase transition temperature ($M_S$) is enhanced, and combined with Table 1 and Equations (1) and (2), the $M_S$ can be expressed in terms of electron concentration.

**Table 2.** Data measured by DSC differential calorimetry scanner.

| | $A_S$(K) | $A_f$(K) | $A_p$(K) | $M_S$(K) | $M_f$(K) | $M_p$(K) | $A_p$–$M_p$(K) | $\Delta H_{Cooling}$ (J/g) | $\Delta H_{Heating}$ (J/g) |
|---|---|---|---|---|---|---|---|---|---|
| $Ni_{50}Mn_{29}Ga_{21}$ Polycrystalline alloy microwire | 286 | 290 | 288 | 273 | 267 | 272 | 16 | 4.67 | 1.83 |
| $Ni_{50}Mn_{25}Ga_{20}Fe_5$ Polycrystalline alloy microwire | 336 | 342 | 339 | 337 | 321 | 332 | 7 | 10.32 | 5.61 |

For ordered heat-treated Ni-Mn-Ga alloy microwires there is the formula:

$$M_S(\text{K}) = 703(e/a) - 5092 \tag{3}$$

For ordered heat-treated Ni-Mn-Ga-Fe alloy microwires there is the formula:

$$M_S(\text{K}) = 705(e/a) - 5106 \tag{4}$$

According to the energy band theory, the crystal structure is periodically stable with a certain value of $e/a$ [22]; when $e/a$ reaches a critical value, the "Fermi surface", the boundary between free electrons and non-electrons in the lattice, reaches the boundary of its Brillouin zone, the free electrons outside the Fermi surface preferentially jump into the Brillouin zone, the free electrons in the original Brillouin zone surge, the lattice internal free distortion to maintain the overall stability of the lattice structure, the original Brillouin zone of high energy free electrons freely combined into a new Brillouin zone, so that the lattice structure with a new period of stability. From the above, it can be seen that Fe has more valence electrons than Ga; combined with Table 1, it can be seen that the $e/a$ of Fe-doped microwires has increased, indicating that Ni-Mn-Ga-Fe microwires are more likely to reach the critical value, and the probability of the Fermi surface coinciding with the Brillouin zone increases [22], and the instability of the lattice structure inside the crystal decreases, and the temperature required to complete MT increases.

In addition to the effect of $e/a$ on MT temperature, cell size and crystal second phase are also the main factors affecting MT temperature. The microwires selected for this experiment were all fine microwires with uniform texture and subject to SE testing, and no precipitation phase was observed in both SEM and XRD; therefore, in addition to $e/a$, the effect of unit cell size on MT temperature needs to be investigated. The apparent macroscopic structure of the microwires did not change as observed by SEM, indicating that the main strain value of the Bain distortion [23] is 0. In the MT process, the inertial surface composed of the self-collaboration effect and the dot-invariant property first undergoes

lattice reformation. As the radius of Fe is smaller than that of Ga, the cell volume of Fe doped microwires decreases, which is in accordance with the aforementioned bring down in the lattice structure instability and microwire density, and the induced phase transition temperature required for the formation of the self-collaborative martensitic state increases in the MT process [22].

The DSC curve data in Table 2 define $M_S$, $M_f$, $A_S$, and $A_f$ as the onset and end temperatures of MT and the onset and end temperatures of inverse phase transformation, respectively; the data from the table confirm that the MT temperatures are increased. In addition to the influence of $e/a$ and the cell size of microwires, the high melting degree of Fe itself is also one of the reasons for the increase of the MT temperature of microwires. Define $A_p$ and $M_p$ as the peak temperature point of martensite phase change and inverse phase change respectively, $A_p$–$M_p$ indicates the heat lag of martensite phase change, which is mainly generated by the friction between newborn martensite and parent phase interface in the MT process, a kind of irreversible heat loss; Table 2 illustrates that Fe doping reduces the thermal hysteresis because the unit cell size of Fe doped microwires decreases and the uniformity of the microwire texture increases, which makes the MT process do less work to overcome the phase interface friction and the elastic strain recoverability of the microwires themselves increases, effectively reducing the martensitic phase transformation thermal hysteresis.

Define $\Delta H_{\text{Cooling}}$, $\Delta H_{\text{Heating}}$ as the martensite phase transition enthalpy and martensite inverse phase transition enthalpy, which are obtained from the integral of the area of the exothermic peak enclosed by ($M_S$–$M_f$) and $M_p$ in the cooling curve and the integral of the area of the heat absorption peak enclosed by ($A_f$–$A_S$) and $A_p$ in the heating curve of Figure 3b, respectively. According to the definition of enthalpy change in thermodynamics, the state quantity $\Delta H$ is only related to the thermodynamic energy change of the MT process [5,24]; As can be seen, Ni-Mn-Ga microwires are mostly of a 5M martensite structure, while Fe doped microwires are mostly of a 7M martensite structure; 7M martensite is more compact than 5M martensite, and the radius of atoms has been reduced, leading to more irregular thermal motion of atoms in the lattice, and the increase of molecular kinetic energy makes the enthalpy change $\Delta H$ increase [24]. It is known from molecular kinetic theory that the increase of molecular kinetic energy marks the increase of MT temperature, which is in accordance with the law of Fe doping to increase MT temperature, shown in Figure 3b. In the SE process, high $\Delta H$ can provide more chemical free energy to provide the driving force for SE deformation recovery, reduce the irrecoverable deformation $\varepsilon_r$ due to phase interface friction, preparation defects, etc.; and the SE test curves for different temperatures are shown in Figure 4.

The Q800 dynamic analyzer is chosen for this experiment to test the SE curves of microwires at different temperatures, hoping that its stress-induced MT process can get a larger $\varepsilon_t$ and smaller $\varepsilon_r$ to give full play to its own elastic nature and explore the SE recovery characteristics of its inelastic deformation; at the same time, it is necessary to avoid the dislocation slip and deformation twinning caused by excessive stress. Figure 4a shows the SE test curves of $Ni_{50}Mn_{29}Ga_{21}$ microwires at different temperatures. First, the SE test should meet the test condition of $T_{\text{test}} > M_S$; second, from Figure 4a, it is found that the total strain of the microwire decreases when $T_{\text{test}} = 315$ K, which indicates that irreversible dislocations occur inside the microwire at this time, and it is not advisable to continue to raise the temperature for the SE test; third, in Figure 4b, when $T_{\text{test}} = 359$ K, the recoverable deformation of the microwire is nearly 100%, the ideal value is reached, and there is no need to further warm up for SE testing. Therefore, combining the MS of the two microwires in Table 2 and in the above analysis, it can be seen that the SE test temperatures of $Ni_{50}Mn_{29}Ga_{21}$ microwire and $Ni_{50}Mn_{25}Ga_{20}Fe_5$ microwire were selected as 301 K, 303 K, 304.5 K, 306 K, 308 K, 310.5 K, 313 K, 315 K and 346 K, 348 K, 350 K, 351.5 K, 353 K, 355 K, 357 K, and 359 K. Among them, $T_{\text{test}} = 310.5$ K in Figure 4c is a typical SE curve of $Ni_{49.9}Mn_{28.6}Ga_{21.5}$ microwire, which is composed of the applied stress-induced MT curve and the unloading stress deformation recovery curve [25]. The stress-induced MT curve is divided into three parts with green vertical lines in the figure, which represent

(I) austenite elastic strain interval, (II) martensite phase transformation interval and (III) residual austenite and nascent martensite elastic deformation and remaining minority stress-induced MT interval; the intersection of the tangent lines between (I) and (II) intervals is the critical stress value of martensite phase transformation, denoted by $\sigma_{M_S}$ [25]. For strain analysis of the unloading stress-deformation recovery curve section, the total, elastic, superelastic, and residual strains can be denoted by $\varepsilon_t$, $\varepsilon_e$, $\varepsilon_{SE}$, and $\varepsilon_r$, respectively, which are delineated in the figure by red, blue, yellow, and purple double arrows; and the inflection point of the elastic strain curve during unloading is the critical stress value of austenite phase transformation, denoted by $\sigma_{M_f}$ [25].

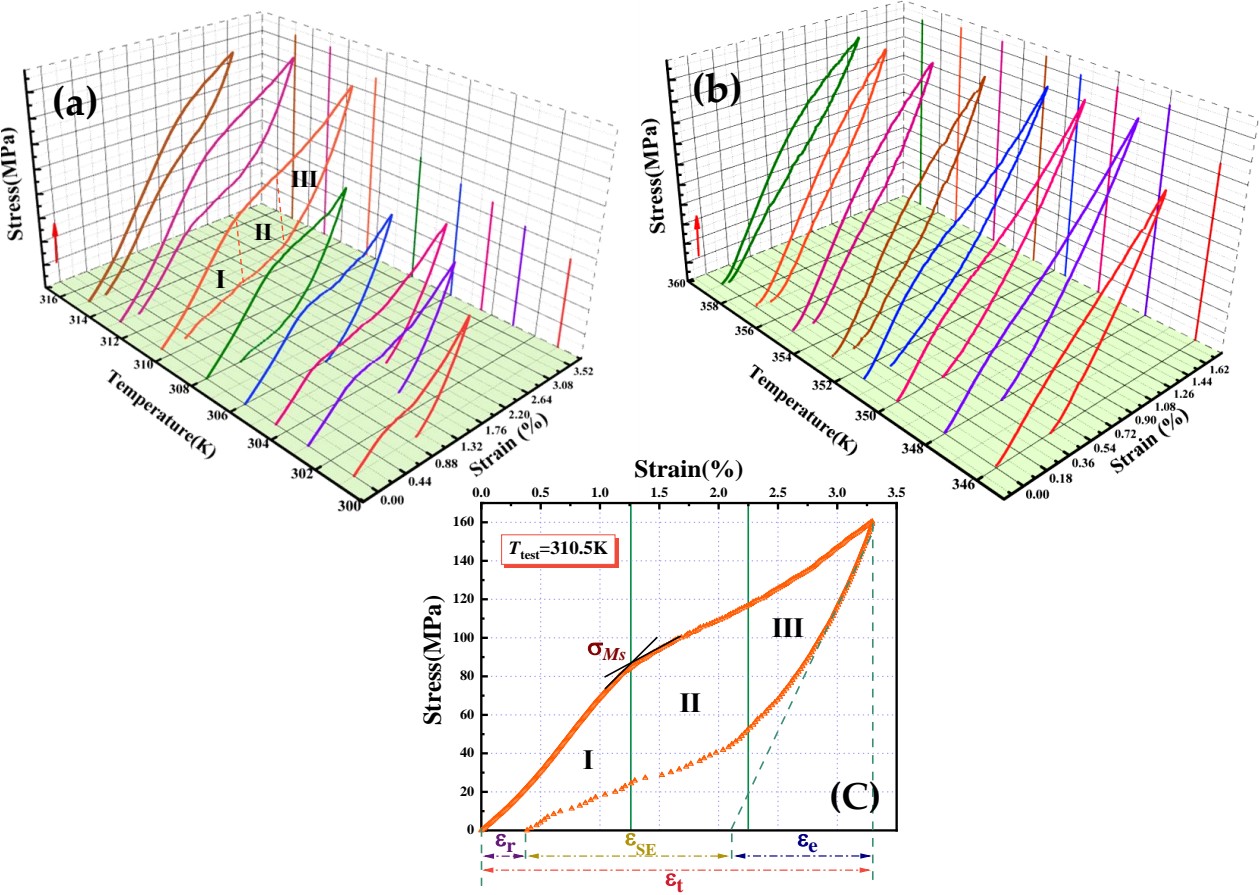

**Figure 4.** (**a**) SE curves of $Ni_{50}Mn_{29}Ga_{21}$ microwire, (**b**) $Ni_{50}Mn_{25}Ga_{20}Fe_5$ microwire SE curve, (**c**) Typical curve of superelasticity.

The area enclosed by the SE curve of $Ni_{50}Mn_{25}Ga_{20}Fe_5$ microwire in Figure 4 is smaller than that of $Ni_{50}Mn_{29}Ga_{21}$ microwire, and the area enclosed by the curve indicates the hysteresis area of the microwire, indicating that the Fe doping makes the microwire consume less energy during the SE process. This is due to the fact that the energy stored in the MT process is released during the unloading stress process, which can better complete the $\varepsilon_e$ and $\varepsilon_{SE}$, and it verifies the previous statement that "high $\Delta H$ can provide more chemical free energy to drive the SE deformation recovery". In addition, the $Ni_{50}Mn_{25}Ga_{20}Fe_5$ microwire has almost no stress plateau in the MT interval, which is similar to the linear superelasticity of (I) and (III) [23]; this indicates that MT occurs almost continuously, which means that Fe doping can refine the microwire grains and stress continuously induces the microwire transformation from austenite to martensite. From the stress-temperature projection images of both microwires in Figure 4, the load stress corresponding to the maximum strain value, when the microwire reaches the maximum strain value, increases continuously with the increase of temperature; the SE curve as a whole shifts toward the large stress direction,

indicating that Fe doping can increase the tensile strength of the microwire. In the process of unloading stress, as the inflection point of the SE curve of Fe doping is not obvious, the tangent tracing points are made one by one, and the critical stress-temperature curves of the two microwires are plotted as Figure 5 by combining the $\sigma_{M_s}$ and $\sigma_{M_f}$ defined previously.

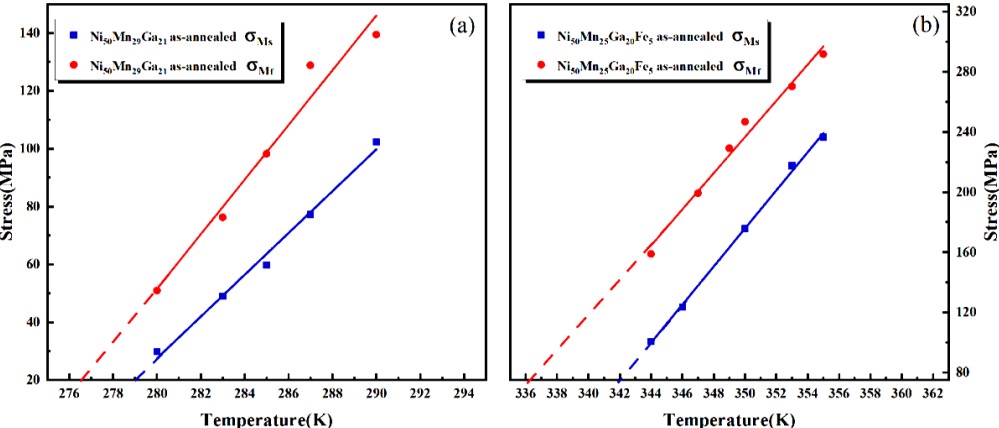

**Figure 5.** (**a**) Critical stress-temperature curves for $Ni_{50}Mn_{29}Ga_{21}$ microwires, (**b**) Critical stress-temperature curves for $Ni_{50}Mn_{25}Ga_{20}Fe_5$ microwires.

Figure 5a shows the critical stress-temperature curve of $Ni_{50}Mn_{29}Ga_{21}$ microwire, and Figure 5b shows the critical stress-temperature curve of $Ni_{50}Mn_{25}Ga_{20}Fe_5$ microwire. From the figure, it can be seen that the critical stress and temperature of both microwires are linearly related to each other, which will be expressed in the following equation.

For $Ni_{50}Mn_{29}Ga_{21}$ microwires, the formula is as follows:

$$\sigma_{M_s} = 7.2 \times (T - 276.2) \tag{5}$$

$$\sigma_{M_f} = 9.5 \times (T - 274.5) \tag{6}$$

For $Ni_{50}Mn_{25}Ga_{20}Fe_5$ microwires the formula is as follows:

$$\sigma_{M_s} = 12.7 \times (T - 336.2) \tag{7}$$

$$\sigma_{M_f} = 12.1 \times (T - 330.3) \tag{8}$$

From Equations (5), (6), (7) and (8), it can be seen that the interval for completing the MT phase transformation in $Ni_{50}Mn_{29}Ga_{21}$ microwires is 274.5~276.2 K, and the interval for completing the MT phase transformation in $Ni_{50}Mn_{29}Ga_{21}$ microwires is 330.3~336.2 K; it indicates that Fe doping can broaden the martensitic phase transformation interval in SE. From the critical stress values in the vertical coordinates of Figure 5, it can be seen that Fe doping increases the critical stress of microwires by 70~200 MPa, which is because Fe doping increases the twin boundaries inside the microwire crystals, resulting in an increase in the critical stress required for stress-induced martensitic phase transformation (SIMT); and the solid solution strengthening of Fe can increase the critical stress of microwires significantly. The slope of the critical stress of the microwire versus temperature, $d\sigma/dT$, indicates the sensitivity of the critical stress of the microwire to the superelasticity temperature, and the Fe doped microwire in the above equation is more sensitive to temperature, which is due to the fact that the Fe doping makes the cell volume inside the microwire lower and the smaller cell volume is more sensitive to the temperature change. Meanwhile, both the above equation and the enthalpy change $\Delta H$ in Table 2 satisfies the Clausius-Clapeyron equation [26,27] ($d\sigma/dT = -\Delta H/T\varepsilon$), which means that the rate of change of critical stress with temperature is proportional to the enthalpy of phase change and inversely proportional to temperature.

The SE strain values in Figure 4 are the focus of the microwire SE study, the strain-temperature relationship curves (the temperature difference is the difference temperature between the test temperature and $M_S$, expressed as $\Delta T$) of Ni$_{50}$Mn$_{29}$Ga$_{21}$ and Ni$_{50}$Mn$_{25}$Ga$_{20}$Fe$_5$ microwires are plotted in Figure 6 according to $\varepsilon_t$, $\varepsilon_e$, $\varepsilon_{SE}$, and $\varepsilon_r$ in Figure 4. Figure 6a represents the total strain versus temperature difference curves of the two microwires, the $\varepsilon_t$ of Ni$_{50}$Mn$_{29}$Ga$_{21}$ microwire reaches the maximum value firstly and then decreases with the increase of temperature, while the Ni$_{50}$Mn$_{25}$Ga$_{20}$Fe$_5$ microwire decreases slightly from the maximum value; this indicates that Ni$_{50}$Mn$_{29}$Ga$_{21}$ microwire still has some SIMT not induced when it is close to the $M_S$ state, as it needs to be warmed up continuously to reach the maximum value; while Ni$_{50}$Mn$_{25}$Ga$_{20}$Fe$_5$ microwire can reach the maximum $\varepsilon_t$ near $M_S$, which is verified with the aforementioned "Fe doping makes the microwires more sensitive to temperature". In addition, because of the grain refinement, increase in grain boundaries, the grain boundaries play a limiting role on the intracellular atomic occupancy, the $\varepsilon_t$ of Ni$_{50}$Mn$_{25}$Ga$_{20}$Fe$_5$ microwire decreases compared to Ni$_{50}$Mn$_{29}$Ga$_{21}$ microwire, with the maximum value decreasing from 3.29% to 1.48%. Nevertheless, the fluctuation of the total strain with increasing temperature difference after reaching the maximum $\varepsilon_t$ was $1.5 \pm 0.3\%$ for Ni$_{50}$Mn$_{29}$Ga$_{21}$ microwires, while it was only $0.5 \pm 0.1\%$ for Ni$_{50}$Mn$_{25}$Ga$_{20}$Fe$_5$ microwires. The decrease of $\varepsilon_t$ is generally considered to be due to the friction between twin boundaries and the irreversible defects caused by multiple temperature increases, so it can be seen that Fe doping can reduce this part of defects through grain refinement, making the utilization of Ni-Mn-Ga-Fe microwires improve.

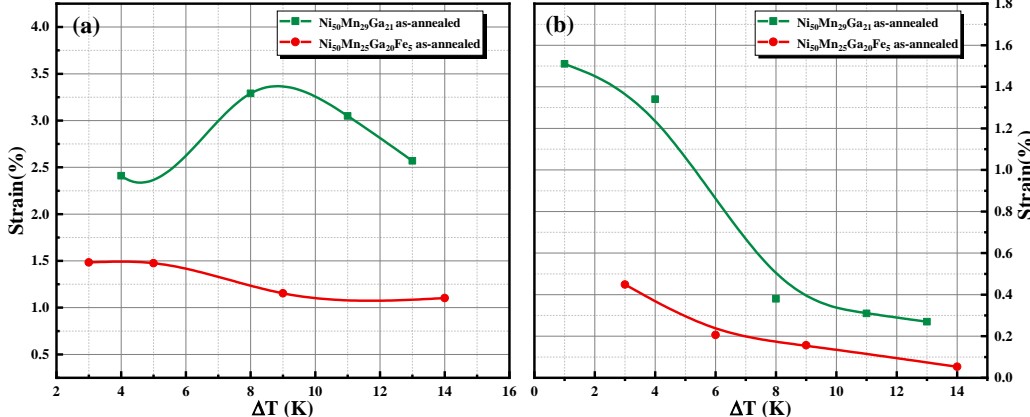

**Figure 6.** (**a**) Total microwire strain-temperature difference curve, (**b**) microwire residual strain-temperature difference curve.

Figure 6b shows the residual strain versus temperature difference curves for the two microwires. It can be seen from the figure that the $\varepsilon_r$ of both microwires decreases continuously from the maximum value with increasing temperature difference. The maximum $\varepsilon_r$ of Ni$_{50}$Mn$_{29}$Ga$_{21}$ microwires is 1.51% and that of the Ni$_{50}$Mn$_{25}$Ga$_{20}$Fe$_5$ microwire is 0.45%. Combined with the SE curve in Figure 4, it can be seen that for the Ni$_{50}$Mn$_{29}$Ga$_{21}$ microwires at $T_{\text{test}}$ of 301K~306K, the microwires only show their own elastic recovery characteristics. Combined with the total strain curve in Figure 6a, it can be seen that as the temperature difference $\Delta T < 8$ K, the microwires themselves increase with temperature, making SIMT complete and $\varepsilon_t$ significantly higher, and the elastic strain curve in Figure 7a, it is known that $\varepsilon_e$ rises rapidly at this time, then $\varepsilon_r = \varepsilon_t - \varepsilon_e$ ($\varepsilon_{SE} \approx 0$) there is a sudden drop process. After $\Delta T > 8$K, Ni$_{50}$Mn$_{29}$Ga$_{21}$ microwires continue the SE test at an elevated temperature, which is accompanied by the generation of $\varepsilon_{SE}$, and even though the defects caused by the SE test will reduce $\varepsilon_e$, $\varepsilon_r$ is only slightly reduced and shows a flat trend; $\varepsilon_r$ of Ni$_{50}$Mn$_{25}$Ga$_{20}$Fe$_5$ microwires shows only a flat trend with temperature, which is also because the microwires can be $M_S$ near the maximum $\varepsilon_t$, both $\varepsilon_e$ and $\varepsilon_{SE}$ of the microwire have been produced near $M_S$. And with the increasing temperature, the complete SE is

approximated at $\Delta T = 14$ K. This is because the Fe doping makes the grain refinement, which can reduce intracellular atomic dislocations and increase the microwire stress-release surface area caused by.

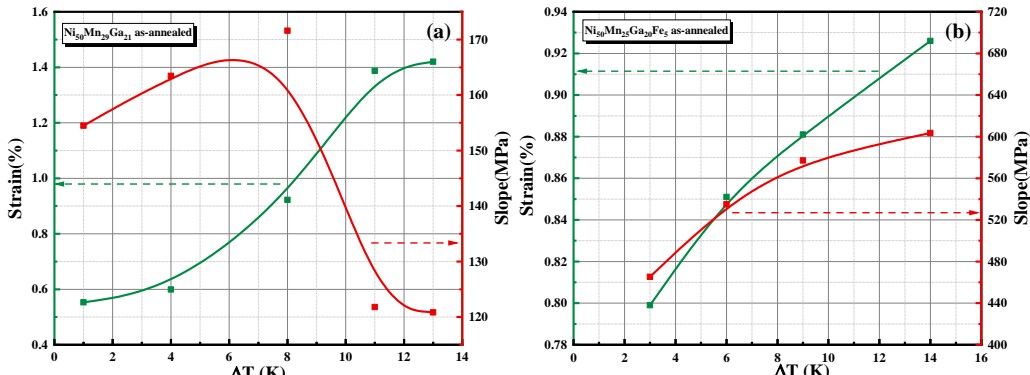

**Figure 7.** (**a**) Relationship curves between elastic strain, elastic modulus and temperature difference for $Ni_{50}Mn_{29}Ga_{21}$ microwire, (**b**) Relationship curves between elastic strain, elastic modulus and temperature difference for $Ni_{50}Mn_{25}Ga_{20}Fe_5$.

Figure 7a shows the $Ni_{50}Mn_{29}Ga_{21}$ microwire elastic strain and elastic modulus versus temperature difference curves. This figure can be divided into leaving and right parts of $\Delta T = 8$ K. Combining with the $Ni_{50}Mn_{29}Ga_{21}$ microwire curve in Figure 6a, it can be seen that even though $\varepsilon_e$ shows an increasing trend in $\Delta T > 8$ K, the elastic modulus and $\varepsilon_t$ drop sharply, indicating that the microwire's own ability to resist elastic deformation decreases significantly when $\Delta T > 8$ K due to obvious defects inside the microwire caused by multiple SE tests. At this time, it is accompanied by the isotropic displacement of the lattice, so the elastic deformation still shows an increasing trend; until the temperature difference reaches 12 K, the microwire defects are basically fixed, and the elastic modulus and $\varepsilon_t$ are approximately stable. Yet Fe doped microwires show an increasing trend in both the modulus of elasticity and $\varepsilon_e$, and the curves are shown in Figure 7b. The modulus of elasticity of this microwire increases with the temperature, but the slope does have a slow decrease, indicating that the microwire in the SE test due to the continuous heating, loading and unloading process of the load and microwire friction will still produce a very small number of defects; and the increase in the temperature of the twin boundary mobility will also affect the modulus of elasticity, but the overall resistance to deformation of the microwire has not been destroyed, and the elastic deformation still indicates an upward trend, in Figure 4, the overall appearance of "linear SE".

The yellow and purple lines in Figure 8 show the recoverable strain, strain recovery rate versus temperature difference curves for the two microwires, respectively. First, the recoverable deformation is expressed as $\varepsilon_{rec} = \varepsilon_e + \varepsilon_{SE}$ [17], and the strain recovery rate is expressed as $\eta = \varepsilon_{rec}/\varepsilon_t$. Figure 8a shows the recoverable strain, strain recovery rate and temperature difference relationship curves of $Ni_{50}Mn_{29}Ga_{21}$ microwires. Combined with the elastic strain versus temperature difference curve in Figure 7a, it can be seen that $\varepsilon_{rec}$ of the microwire is only provided by $\varepsilon_e$ when $\Delta T < 8$ K; when $\Delta T > 8$ K, $\varepsilon_{rec}$ does not decrease significantly, indicating that $\varepsilon_{SE}$ of the microwire dominates at this time, and the maximum strain recovery rate of the microwire is stable at 90%, as seen from its strain recovery rate curve, which also indicates that $Ni_{50}Mn_{29}Ga_{21}$ microwires produced irreversible defects in the SE test. Figure 8b shows the recoverable strain and strain recovery rate versus temperature difference curves of the $Ni_{50}Mn_{25}Ga_{20}Fe_5$ microwire. Although the recoverable deformation of the microwire is fluctuating, the recoverable strain fluctuation interval is only in the range of $0.1 \pm 0.05\%$ compared with Figure 8a, which can be approximately considered that the recoverable strain remains stable, which is one of the reasons for the approximately linear SE curve in Figure 4b. Simultaneously, the recoverable strain curve presents characteristics that may be due to the interval fluctuations caused by

the fact that *e/a* reaches a certain critical value at SIMT and the lattice structure needs to exist stably with a new cycle. The corresponding strain recovery is then increasing from 69% to approximately 100%, indicating that Fe doping can still affect the change in $\eta$ with a small change in $\varepsilon_{rec}$ in the presence of reduced microwire $\varepsilon_t$. Even though $\varepsilon_{rec}$ reaches the minimum value at $\Delta T = 9$ K, the ratio still shows an increasing trend at this time when $\varepsilon_t$ decreases by 0.7%, which fully demonstrates the improving effect of Fe doping on the strain recovery rate.

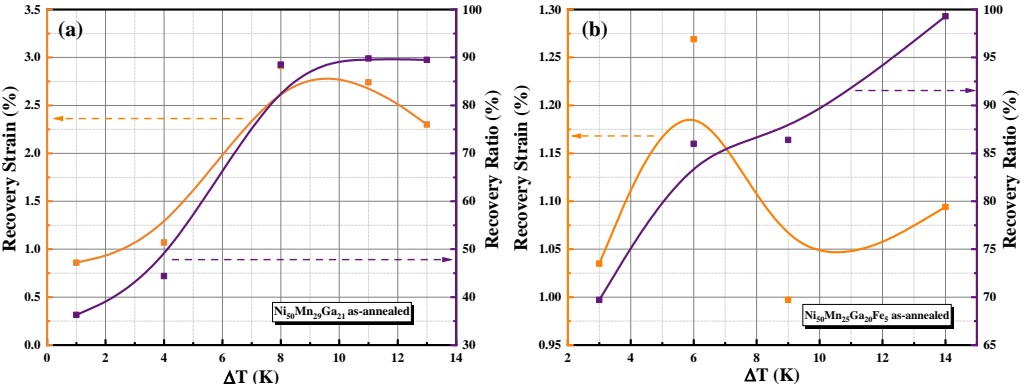

**Figure 8.** (**a**) Relationship curves of recoverable strain, strain recovery rate and temperature difference for $Ni_{50}Mn_{29}Ga_{21}$ microwires, (**b**) Relationship curves of recoverable strain, strain recovery rate and temperature difference for $Ni_{50}Mn_{25}Ga_{20}Fe_5$ microwires.

Figure 9 shows the relationship curves of the SE strain and applied strain for both microwires. SE strain values of both microwires first reach a maximum and then decrease with increasing applied strain. This is due to the increasing $\varepsilon_{SE}$ in the process of SIMT with increasing temperature until the completion of the induced MT; and the stable presence of high temperature austenite leads to the reduction of the martensitic state, which indirectly leads to the reduction of the SE strain. The maximum SE strain value of $Ni_{50}Mn_{29}Ga_{21}$ microwire is 1.35%, and the maximum SE strain value of $Ni_{50}Mn_{25}Ga_{20}Fe_5$ microwire is 0.43%; while the Fe doping can reach the maximum SE strain value at a smaller applied strain, indicating that the $Ni_{50}Mn_{25}Ga_{20}Fe_5$ microwire has a regular lattice structure and small unit cell size, which is easy to SE testing.

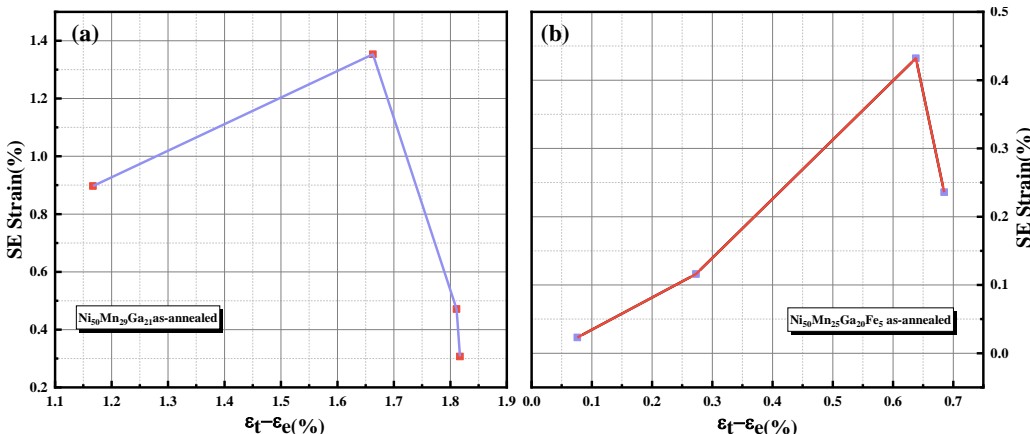

**Figure 9.** (**a**) SE strain-applied strain curves of $Ni_{50}Mn_{29}Ga_{21}$ microwire, (**b**) SE strain-applied strain curves of $Ni_{50}Mn_{25}Ga_{20}Fe_5$ microwire.

## 4. Conclusions

Fe doping refines the internal grains of the microwires, increases the number of free electrons in the lattice, and increases the electron concentration of the microwires. In the

martensitic state, the increase in the number of free electrons leads to an increase in the number of new Brillouin zones, a decrease in the atomic radius, and a significant twin boundary, exhibiting a single 7M modulated martensitic structure superior to that of Ni-Mn-Ga microwires (5M and 7M mixed modulated martensite); $M_S$ also increases by nearly 70 K due to the decrease in cell size and the increase in lattice structure stability; $\Delta H$ increases significantly due to the irregular motion caused by the increase in atoms within the lattice, which can provide more chemical free energy to drive SE deformation recovery and reduce $\varepsilon_r$ to reach full SE.

In the SE test, the tensile strength of Ni-Mn-Ga-Fe microwires is increased due to the hard solid solution strength of Fe itself, and the single structural stability of the microwires results in significantly higher critical stress values for SIMT, with the critical stress exhibiting greater sensitivity to temperature. Both together lead to a complete release of Ni-Mn-Ga-Fe microwires during the SE test, corresponding to an increase in the martensitic phase transition interval and a 10% increase in the recoverable strain rate of the microwires compared to Ni-Mn-Ga microwires, reaching the maximum SE value at low applied strain.

**Author Contributions:** Conceptualization, Y.L., Z.L.; methodology and data curation, Y.L., Z.L.; writing—original draft preparation, Y.L., Z.L.; supervision, H.S., J.L., J.S. All authors have read and agreed to the published version of the manuscript.

**Funding:** This work was supported by National Natural Science Foundation of China Projects (Grant No. 51701099), Natural Science Foundation of Heilongjiang Province (Grant No. LH2019E091) and 2020 Heilongjiang Provincial Undergraduate University Basic Research Business Fund for Young Innovative Talents Project (Grant No. 135509218).

**Conflicts of Interest:** The authors declare that they have no conflict of interest.

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
