# Peer review of "Martensitic Transition and Superelasticity of Ordered Heat Treatment Ni-Mn-Ga-Fe Microwires"

_metals, doi:10.3390/met12091546_

Round 1

Reviewer 1 Report

The article entitled "Martensitic transition and Superelasticity of ordered heat treatment Ni-Mn-Ga-Fe microwires" is an application paper that compares Ni-Mn-Ga and Ni-Mn-Ga-Fe microwires regarding general SMA properties.

The article is interesting but needs minor revision before it can be published in Metals.

1. The abstract needs revision in English. There is a significant number of repetitive connections of phrases via "and".
This is only grammatically correct, but not a good style that is easy to read.

2. On page 4 line 136:
"Meanwhile, the increase in electron concentration is due to the fact that the valence electron number 3d64s2 of Fe is more active
than the valence electron number 4s24p1 of Ga."

How can a number be more active than another one?

3. Page 5: "As can be seen from the figure, Fe doping makes the MT temperature shift to right, indicating that the martensite phase transition temperature (MS) is enhanced, and ..."

There is no left or right in science. Instead, a temperature can be shifted either to higher or lower values. A transition temperature can be enhanced?

4. page 5, line 167:  "....the instability inside the crystal increases,"   
what instability? Exact expression required.

5. A general point: The authors do not compare their results with those of similar articles/materials, they put only the found facts, which limits
the interest to the readers. This is a major weakness of the manuscript.

Reviewer 2 Report

The manuscript describes the effects of Fe doping on the microstructure, martensitic transformation and functional property of Ni-Mn-Ga(Fe)  microwires. The authors present some novel and significant results. This work represents a substantial contribution in the field of advanced FSMAs.

 However, I would like to suggest a few corrections, which I believe, can improve the manuscript.

 1.     Many sentences are very long and hard to read, e.g., " The preparation of master alloy ingots and microwires of Ni-Mn-Ga and Ni-Mn-Ga-Fe  was completed by high vacuum electric furnace melt melting furnace and melt drawing liquid forming equipment…..,. …." (in the abstract). ‘The Ni-Mn-Ga-Fe microwire grain size l=1.264±0.195μm, which is smaller than that of Ni-Mn-Ga microwire grain l=2.526±0.150μm, when… Because the atomic radius of Fe 128 is smaller than that of Ga (rFe=0.172nm<rGa=0.181nm), Fe doping changes the first nearest 129 neighbor of the internal lattice of the microwire from …..(in the Results and Discussion ),  etc. The authors should try to focus on the key info and break these sentences.

2.     The text should be carefully double-checked. There are many typing mistakes.

3.     I believe that it is enough to describe the standard research methods in “ 2. Experimental Procedure”.  It is superfluous information in the abstract.

4.     Pg.2 line 53. “In this experiment, the fourth group element Fe was selected because it is an austenitic stabilizing element with….” Thus, the authors have expected a decrease in martensitic transformation temperatures in Ni-Mn-Ga-Fe microwires compared to Ni-Mn-Ga ones. Indeed, the doping of Fe results in decrease the martensitic transformation temperatures in TiNi SMA and CoNiGa FSMAs [for example, X. F. Dai….. APPLIED PHYSICS LETTERS 87, 112504 2005]. Why does the Fe become the martensite stabilizing element and results in increase the martensitic transformation temperatures in NiMnGa-based alloy?

5.     Figure 3 (a). What are the test temperatures of XRD study?

6.     Figure 3 (b) and Table 2. Temperatures of MT have determined as Ms =344 K and Af= 342 K in Ni50Mn25Ga20Fe5  microwire. It is impossible. The reverse transformation cannot finish at lower temperature Af= 342 K then the forward transformation begins (Ms =344 K).  How authors to define the ∆HHeating for Ni50Mn25Ga20Fe5  microwire? As follows from the Figure 3 b, the ∆HHeating should be much less than ∆HCooling, considering the location of the As and Af points.

7.     The parameters of SE curves are very hard to distinguish in Figure 4(a).

8.     Why the authors choose the last test temperature T=315 K for Ni-Mn-Ga microwire  and T=359 K for Ni-Mn-Ga-Fe microwire during study SE response? The authors should specify this moment.

9.     Define what is temperature difference ΔT in Figures 6, 7 and 8.

10. When the authors discuss the increase in the enthalpy change of transformation ∆H with Fe doping, they do not take into account the formation of a different martensite structure 5 M and 7 M in Ni-Mn-Ga and Ni-Mn-Ga-Fe microwires respectively. Why?

 The results obtained are sound and useful for the scientific community; however, modification based on provided comments are seems to be necessary, and the manuscript should be accepted only after a major revision.

Round 2

Reviewer 2 Report

The authors have made all the necessary changes to the manuscript in accordance with the reviewer’s comments. The article can be accepted in present form. It is only one remark.

Pg 6, line 204. “As can be seen before, Ni-Mn-Ga microwires are mostly 5M martensite structure, while Fe doped microwires are mostly 7M martensite structure, 7M martensite is more compact than 5M martensite, and the radius of atoms has been reduced, leading to more irregular thermal motion of atoms in the lattice, and the increase of molecular kinetic energy makes the enthalpy change ∆H increases [24]

The reviewer thinks that Ref. [24] is an error. In Ref. [24] the mechanical properties of a Ni54Mn25Ga21 (at.%) high-temperature shape memory alloy undergoing transformation to tetragonal non-modulated martensite has been  studied, and there are no data about the transformation enthalpy change.